# Exploring the Antibiotic Resistance Profile of Clinical *Klebsiella pneumoniae* Isolates in Portugal

**DOI:** 10.3390/antibiotics11111613

**Published:** 2022-11-13

**Authors:** Ricardo Oliveira, Joana Castro, Sónia Silva, Hugo Oliveira, Maria José Saavedra, Nuno Filipe Azevedo, Carina Almeida

**Affiliations:** 1INIAV—National Institute for Agrarian and Veterinarian Research, Rua dos Lagidos, 4485-655 Vila do Conde, Portugal; 2LEPABE—Laboratory for Process Engineering, Environment, Biotechnology and Energy, Faculty of Engineering, University of Porto, Rua Dr. Roberto Frias, 4200-465 Porto, Portugal; 3AliCE—Associate Laboratory in Chemical Engineering, Faculty of Engineering, University of Porto, Rua Dr. Roberto Frias, 4200-465 Porto, Portugal; 4Center of Biological Engineering (CEB), Campus de Gualtar, University of Minho, 4710-057 Braga, Portugal; 5LABBELS—Associate Laboratory, Braga/Guimarães, 4710-057 Braga, Portugal; 6Department of Veterinary Sciences, Laboratory Medical Microbiology, Antimicrobials, Biocides and Biofilms Unit, University of Trás-os-Montes and Alto Douro, 5000-801 Vila Real, Portugal; 7CITAB—Center for the Research and Technology Agro-Environmental and Biological Sciences, University of Trás-os-Montes and Alto Douro, 5000-801 Vila Real, Portugal; 8Inov4Agro—Associate Laboratory for Innovation, Capacity Building and Sustainability in Agri-Food Production, University of Trás-os-Montes and Alto Douro, 5000-801 Vila Real, Portugal

**Keywords:** *Klebsiella pneumoniae*, β-lactams, antimicrobial susceptibility, carbapenemases, extended-spectrum β-lactamases (ESBL), AmpC β-lactamases

## Abstract

While antibiotic resistance is rising to dangerously high levels, resistance mechanisms are spreading globally among diverse bacterial species. The emergence of antibiotic-resistant *Klebsiella pneumoniae*, mainly due to the production of antibiotic-inactivating enzymes, is currently responsible for most treatment failures, threatening the effectiveness of classes of antibiotics used for decades. This study assessed the presence of genetic determinants of β-lactam resistance in 102 multi-drug resistant (MDR) *K. pneumoniae* isolates from patients admitted to two central hospitals in northern Portugal from 2010 to 2020. Antimicrobial susceptibility testing revealed a high rate (>90%) of resistance to most β-lactam antibiotics, except for carbapenems and cephamycins, which showed antimicrobial susceptibility rates in the range of 23.5–34.3% and 40.2–68.6%, respectively. A diverse pool of β-lactam resistance genetic determinants, including carbapenemases- (i.e., *bla*_KPC-like_ and *bla*_OXA-48-like_), extended-spectrum β-lactamases (ESBL; i.e., *bla*_TEM-like_, *bla*_CTX-M-like_ and *bla*_SHV-like_), and AmpC β-lactamases-coding genes (i.e., *bla*_CMY-2-like_ and *bla*_DHA-like_) were found in most *K. pneumoniae* isolates. *bla*_KPC-like_ (72.5%) and ESBL genes (37.3–74.5%) were the most detected, with approximately 80% of *K. pneumoniae* isolates presenting two or more resistance genes. As the optimal treatment of β-lactamase-producing *K. pneumoniae* infections remains problematic, the high co-occurrence of multiple β-lactam resistance genes must be seen as a serious warning of the problem of antimicrobial resistance.

## 1. Introduction

*Klebsiella pneumoniae* is a gram-negative opportunistic pathogen capable of colonizing, invading, and causing infections in diverse anatomical sites of the human body [1,2]. Unfortunately, over the last decades, the indiscriminate use of antibiotics in human health and agriculture has contributed to the emergence of the antibiotic resistance problem [3]. Consequently, the growing resistance of *K. pneumoniae* infections has become a burden for clinicians. β-lactam antibiotics, including penicillin derivatives, cephalosporins, monobactams, and carbapenems, are the most widely used class of drugs in the treatment of bacterial infections [4]. For this reason, the uncontrolled emergence of resistance mechanisms against antibiotics in this family is particularly frightening. Several mechanisms of antibiotic inactivation used by different microorganisms have already been identified, but the production of antibiotic-inactivating enzymes is the most common mechanism [5]. In the case of β-lactams, these enzymes are called β-lactamases, including carbapenemases (CPN), extended-spectrum β-lactamases (ESBL), and Ambler class C cephalosporinases (AmpC) [6].

Resistance to carbapenems is a major concern in human health. CPN may confer resistance to virtually all β-lactams, a reason why this class of β-lactam is considered a highly effective antibiotics in the treatment of serious and life-threatening bacterial infections [7]. They are poorly inhibited by classical β-lactam inhibitors, except for a subgroup called metallo-β-lactamases (MBLs) [8]. It is important to note that *K. pneumoniae* is the most reported carbapenemase-producing bacteria, while the KPC-type carbapenemase, an Ambler class A β-lactamase, is the most frequently associated [8,9]. Other relevant CPN enzymes are MBLs, Ambler class B β-lactamases that include Imipenem-resistant *Pseudomonas* (IMP)-carbapenemases and Verona integron-encoded metallo-β-lactamase (VIM) types, New Delhi metallo-β-lactamase type 1 (NDM-1), and the class D carbapenemases known as OXA-type β-lactamases [8,10].

ESBLs are a group of plasmid-mediated enzymes belonging to Ambler class A of β-lactamases, which can hydrolyze most penicillins, monobactams, and cephalosporins, and hydrolyze third- and fourth-generation cephalosporins classified as critically important antimicrobials. However, they are not able to degrade cephamycins and carbapenems and are inhibited by β-lactamase inhibitors (e.g., clavulanic acid, sulbactam, and tazobactam) and diazabicyclooctanones (e.g., avibactam) [11,12]. From a clinical point of view, the predominant families of ESBL are CTX-M-like, TEM-like, and SHV-like [13].

AmpC-type cephalosporinases are Ambler class C β-lactamases that can hydrolyze penicillins, monobactams, and first-, second-, and third-generation cephalosporins (cephamycins as well as to oxyimino-β-lactams), but neither fourth-generation cephalosporins (e.g., cefepime) or carbapenems. Furthermore, they are poorly inhibited by classical ESBL inhibitors [14]. The appearance of cephalosporinases in bacterial species lacking chromosomal AmpC genes, such as *K. pneumoniae*, is due to the so-called plasmid-mediated AmpC β-lactamase genes. There are several types of mobile AmpC genes identified, and the most widely disseminated are the CMY-2-like and the inducible DHA-like β-lactamases [13]. Although the acquired AmpC β-lactamase has been spread widely, its overall frequency has remained far below that of ESBLs [15].

Considering that the development of antibiotic resistance mechanisms requires the presence of specific resistance genes, genotypic characterization is a valuable tool to identify and/or confirm potential resistance mechanisms and provide the most effective treatment [15]. Moreover, the study of the distribution and prevalence of antibiotic resistance genes is pivotal from an epidemiological point of view in order to understand their dissemination in an increasingly connected world [16]. In this study, we first analyzed the antimicrobial susceptibility through the VITEK 2 compact system complemented with the disk diffusion method, and then, we used four multiplex PCR protocols capable of detecting the most commonly acquired β-lactamase genes of CPN (*bla*_KPC-like_, *bla*_OXA-48-like_, *bla*_OXA-23-like_, *bla*_NDM-1-like_, *bla*_VIM-like_, and *bla*_IMP-like_), ESBL (*bla*_SHV-like_, *bla*_TEM-like_, and *bla*_CTX-M-like_), and AmpC (*bla*_CMY-2-like_ and *bla*_DHA-like_) families, to characterize 102 MDR *K. pneumoniae* isolates from patients admitted in two central hospitals in northern Portugal.

## 2. Results

### 2.1. Distribution of the K. pneumoniae Isolates

In total, 102 *K. pneumoniae* isolates collected by the laboratories from two Portuguese hospitals between 2010 and 2020 were used during this work. Strains were isolated from patients admitted to different areas of health care within the hospitals. Despite the source samples being diverse, the most significant was urine (39.2%), rectal swabs (15.7%), blood cultures (12.7%), and sputum (8.8%). Figure 1 shows the source of all isolates included in this work.

To facilitate further analysis, the isolates were divided into five main groups according to sample source: urine (39.2%), rectal swabs (15.7%), respiratory-associated infections (15.7%) (including sputum, bronchial aspirates, and bronchoalveolar aspirates), blood cultures (12.7%), and other samples (16.7%).

### 2.2. Resistance Profiles of K. pneumoniae Isolates to β-Lactam Antibiotics

The susceptibility rates of *K. pneumoniae* isolates for each β-lactam tested (Table 1) are shown in Figure 2. Ertapenem (ERT) was effective (i.e., susceptible profile) against 24/102 (23.5%) isolates, imipenem (IMI) was effective against 33/102 (32.4%) isolates, meropenem (MER) was effective against 35/102 (34.3%) isolates, and doripenem (DOR) was effective against 34/102 (33.3%) isolates.

The susceptibility rates of the other β-lactams were lower or even null for most isolates, with resistance rates >90% for all β-lactams tested other than the carbapenems. Only cephamycins, cefoxitin (CXI), and cefotetan (CTT) exhibited greater effectiveness (40.2% and 68.6%, respectively) against *K. pneumoniae* isolates. Furthermore, 21 (20.6%) *K. pneumoniae* isolates showed a complete non-susceptibility profile to all β-lactams tested, including carbapenems.

Based on individual antimicrobial profiles, each isolate was classified as a potential producer of CPN, ESBL, or AmpC β-lactamases according to the criteria recommended by EUCAST for the detection of resistance mechanisms and described in the “Materials and methods” (see Appendix A). Notably, all *K. pneumoniae* isolates showed a potential ESBL-producing phenotype, while 10 (9.8%) isolates had a characteristic phenotype of both AmpC- and/or ESBL-producing bacteria. In addition, 31 (30.4%) isolates exhibited a resistance phenotype that can be classified as CPN- and/or ESBL-producing isolate, and 50 (49.0%) isolates showed a resistance phenotype that can be classified as CPN-, AmpC-, and/or ESBL-producing isolate.

### 2.3. Screening of Acquired β-Lactam Resistance Genes

Prior to the experiments, four multiplex PCRs were designed and optimized for the detection of the most commonly acquired β-lactam resistance genes using primers described in the literature (Table 2). Appendix A shows representative sequences of each of the 11 *bla* gene families of interest with the binding site of the primer pairs used for PCR detection. 

The 102 *K. pneumoniae* clinical isolates were then tested for the presence of the resistance genes according to the PCR methodology described in “Materials and methods”. Figure 3 shows the amplification products of the control strains separated on 1.5% (*w/v*) agarose gel.

The profile of resistance genes of each *K. pneumoniae* isolate can be found in Appendix A. In total, *bla*_KPC-like_ was detected in 74/102 (72.5%) *K. pneumoniae* isolates, while *bla*_OXA-48-like_ was detected only in three isolates (2.9%). No *K. pneumoniae* isolates harbored *bla*_OXA-23-like_ or MBL genes. For the ESBL genes, 76/102 (74.5%) were positive for the presence of *bla*_SHV-like_, 38/102 (37.3%) for the presence of *bla*_CTX-M-like_, and 47/102 (46.1%) for the presence of *bla*_TEM-like_. Among the AmpC genes, 4/102 (3.9%) isolates were positive for *bla*_CMY-2-like_, and 4/102 (3.9%) isolates were positive for *bla*_DHA-like_. *bla*_KPC-like_ was most frequently detected in isolates from urine (32.4%), followed by rectal swabs (14.7%), respiratory-associated infections (8.8%), and blood (7.8%). The isolates positive for *bla*_OXA-48-like_ were only associated with urine (2.0%), and respiratory (1.0%) isolates. Regarding ESBL genes, the distribution by sample sources is wider. *bla*_TEM-like_ was detected in the highest percentage in urine, followed by respiratory-associated infections, other samples, blood culture, and rectal swabs. The most prevalent ESBL gene (*bla*_SHV-like_) was most associated with urine, followed by respiratory-associated infections, other samples, rectal swabs, and blood cultures. In turn, *bla*_CTX-M-like_ was most detected in murine, followed by respiratory-associated infections, blood cultures, other samples, and rectal swabs. *bla*_CMY-2-like_ and *bla*_DHA-like_ were detected in 2.0% of urine isolates and 1.0% of respiratory-associated infections and other isolates. This data can be seen in Figure 4A.

In Figure 4B, it is possible to analyze the relative frequency of each antimicrobial resistance (AMR) gene by type of sample. Within the urine isolates (*n*=40), *bla*_KPC-like_ was most frequent (0.83/1), followed by *bla*_SHV-like_ (0.68/1), *bla*_TEM-like_ (0.45/1), and *bla*_CTX-M-like_ (0.43/1). In isolates from respiratory-associated infections (*n*=16), *bla*_SHV-like_ (0.94/1) was the most prevalent, followed by *bla*_TEM-like_ (0.63/1), *bla*_KPC-like_ (0.56/1), and *bla*_CTX-M-like_ (0.50/1). In rectal swabs (*n*=16), *bla*_KPC-like_ was most prevalent (0.94/1), followed by *bla*_SHV-like_ (0.69/1) and *bla*_TEM-like_ (0.31/1. In blood cultures (*n*=13), *bla*_SHV-like_ was most prevalent (0.69/1), followed by *bla*_KPC-like_ (0.62/1), *bla*_TEM-like_ (0.46/1), and *bla*_CTX-M-like_ (0.46/1). Within the isolates from other samples (*n*=17), *bla*_SHV-like_ was the most frequent (0.88/1), followed by *bla*_KPC-like_ (0.56/1*bla*_TEM-like_ (0.50/1) and *bla*_CTX-M-like_ (0.31/1).

Only one (1.0%) isolate did not provide any positive result for all the genes evaluated, with all the other isolates having at least one acquired β-lactam resistance gene. In total, 19/102 (18.6%) revealed the presence of one, 35/102 (34.3%) revealed the presence of two, 32/102 (31.4%) revealed the presence of three, 14/102 (13.7%) revealed the presence of four, and one (1.0%) isolate revealed the presence of five acquired β-lactam resistance genes.

## 3. Discussion

*K. pneumoniae* belongs to the WHO’s list of antibiotic-resistant pathogens of critical priority that require the development of new antibiotics to combat them [24]. This alarming superbug is also part of the group of pathogens called ESKAPE (*Enterococcus faecium*, *Staphylococcus aureus*, *Clostridium difficile*, *Acinetobacter baumannii*, *Pseudomonas aeruginosa, and* all *Enterobacteriaceae*) normally described for their ability to “escape” from the general antimicrobial therapy [25]. Furthermore, *K. pneumoniae* is a well-known nosocomial pathogen that, in recent years, has been established as an MDR and pandrug-resistant problem [26]. Therefore, understanding the genetic determinants involved in AMR in this bacterial species is important from a clinical and epidemiological point of view [27].

The MDR *K. pneumoniae* isolates used in this study came from a wide range of samples (Figure 1). The main sample sources were urine (39.2%), rectal swabs (15.7%), respiratory-associated infections (15.7%; including sputum, bronchial aspirates, and bronchoalveolar aspirates), and blood (12.7%). Urine represents almost 40% of the origin of the isolates, confirming the increasing trend of MDR *K. pneumoniae* in urinary tract infections (UTIs) [1,28,29]. Furthermore, rectal swabs, respiratory-associated infections, and blood also represented an important source of MDR *K.* pneumoniae, as already reported in several other studies [1,30].

Antimicrobial susceptibility tests revealed a small variety of β-lactam treatment options for the MDR *K. pneumoniae* isolates from Portuguese hospitals (Figure 2). Since this study started from a pool of antibiotic-resistant clinical isolates, high-resistance rates were expected in the susceptibility tests. Therefore, the resistance percentages reported here should not be seen as the prevalence/frequency rates or compared with other studies but as molecular epidemiology data of nosocomial MDR *K. pneumoniae* in Portugal. Still, the high diversity of genetic determinants found in MDR *K. pneumoniae* isolated from two central hospitals confirms that β-lactam efficacy might be seriously threatened [31,32]. Recently, high resistance to β-lactams, including penicillins (60.5%), cephalosporins (66.9%), and carbapenems (60.5%) was also reported among a large collection of *K. pneumoniae* from Portuguese hospitals [33]. In other countries of the European Union/European Economic Area (EU/EEA), the AMR among *K. pneumoniae* isolates is also a serious problem. According to the ECDC’s annual epidemiological report, the AMR percentage for third-generation cephalosporins and carbapenems was 33.9% and 10.0%, respectively, in 2020 [34]. Furthermore, the carbapenem resistance was almost always combined with resistance to several other key antimicrobial groups, leading to a severely limited range of treatment options for *K. pneumoniae*. The highest percentages of carbapenem resistance (>10%) were observed in south and south-eastern European countries. In Portugal, there has been an increasing annual trend of carbapenem resistance among *K. pneumoniae* since 2016 (5.2%% in 2016, 8.6% in 2017, 11.7% in 2018, 10.9% in 2019, and 11.6% in 2020), exceeding the overall prevalence for Europe [34]. This trend is also observed in different studies conducted in other countries, such as China and the United States, with prevalence values reaching close to 15% or 25%, respectively [35,36]. Even so, carbapenems remain the most effective β-lactam antibiotics for most species, so they should be rationally used for the treatment of severe cases involving MDR bacteria, for which traditional β-lactams are not effective [37]. Alternately, the results suggest that cephamycins may be an alternative to carbapenems for the treatment of *K. pneumoniae* infections. Although cephamycins are very similar to cephalosporins and often included in the 2nd generation classification, they are resistant to degradation by ESBL enzymes, and it seems, to a certain extent, to CPN enzymes (especially CTT) [38]. However, they are reported not to be effective against AmpC cephalosporinases and porin mutations, and as noted, most cephamycin-resistant isolates in this work revealed AmpC genes [39].

The classification of potential ESBL- or AmpC-producing species was based on the resistance phenotype to both cefotaxime (CTA) and ceftazidime (CTZ), plus resistance to CXI in the AmpC-producing species. In this way, a phenotype resistant to CTA and CTZ and susceptible to CXI resulted in a classification of potential ESBL-producing bacteria. However, although a CTA, CTZ, and CXI-resistant phenotype corresponds to a potential AmpC-producing isolate, the possibility that it is also an ESBL producer cannot be discarded. Therefore, a phenotype resistant to CTA, CTZ, and CXI was classified as a potential AmpC- and/or ESBL-producing bacteria. Furthermore, it is important to note that for an isolate to be a potential CPN producer, it does not need to have an MER-resistant phenotype. According to the EUCAST criteria, a phenotypic result to MER <28 mm and PIT-resistant with disk diffusion test can be a potential CPN producer. For this reason, some *K. pneumoniae* isolates susceptible to MER (≥22 mm with disk diffusion) may be potential carriers of CPN genes. Indeed, 16 *K. pneumoniae* isolates presented an MER susceptible phenotype (Figure 2) but were classified as potential CPN-producing isolates (Supplementary Material Appendix A). The classification of resistance mechanisms based only on direct evaluation of susceptibility phenotype to individual antimicrobial agents can be a simple way to categorize the resistance of clinical isolates, but it can be ambiguous. Phenotype assessment could be improved by using combinatorial testing of different antimicrobial agents to detect the specific production of β-lactamase classes, for example, by using a combination disk test, double-disk synergy test, or biochemical tests as recommended by EUCAST [15]. Still, PCR screening for acquired antibiotic resistance genes appears to be a more direct way of assessing the resistance potential of clinical isolates.

In general, ESBLs and KPC carbapenemase genes were the most detected (Figure 4), confirming their dissemination in MDR *K. pneumoniae* isolates as verified in several other studies [33,40,41,42,43,44,45,46]. KPC enzymes were originally identified in *K. pneumoniae* which justifies its high prevalence in cases of resistance to carbapenems in this species. However, this type of resistance gene has emerged in other bacterial species, such as *E. coli* and *E. cloacae* isolates [46]. In Europe, *bla*_KPC-like_ (mainly KPC-2 and KPC-3) has already been detected in isolates from Austria, Belgium, the Czech Republic, Germany, Greece, Italy, Romania, and the UK, demonstrating its wide dissemination [47]. In a European survey on carbapenemase-producing *Enterobacteriaceae*, *bla*_KPC_ (18.9%) was the major resistance mechanism reported in clinical isolates from Italy, Greece, Portugal, and Israel [48]. In other Portuguese studies, *bla*_KPC_ genes were also the most detected in clinical *K. pneumoniae* isolates collected in the same time period of our study [33,45,49]. The highest percentage of *bla*_KPC-like_ positive isolates were from urine, which confirms the high prevalence of this pathogen in UTIs [1,29]. In turn, OXA-48 is a carbapenemase that is endemic in North Africa, the Middle East, India, and some European countries such as Turkey, Spain, France, Belgium, the Czech Republic, and Malta [46,48]. However, it has also been detected in *K. pneumoniae* collected from Portuguese hospitals in other studies [33,45]. Recent outbreaks of OXA-48-producing *K. pneumoniae* in Spain, Turkey, and Greece have revealed a serious risk of these strains to human health since they present higher virulence, while laboratory detection is difficult due to their weak capacity to hydrolyze carbapenems [50,51,52]. Nonetheless, in our study, OXA-48-producing *K. pneumoniae* was detected in a low percentage of the *K. pneumoniae* isolates. Regarding MBL, no associated-resistance genes were detected in the isolates. However, the presence of *bla*_VIM-like_ and *bla*_NDM-1_ genes was previously seen in *K. pneumoniae* clinical isolates in Portugal [53,54]. ESBL genes were the most detected in MDR *K. pneumoniae* isolates, demonstrating its high dissemination in Portugal, as is verified in the rest of Europe [55]. Moreover, the overall prevalence of *bla*_SHV-like_ was higher than *bla*_KPC-like_. The high percentage of all ESBL types is directly related to the high percentage of resistance to β-lactams other than the carbapenems, including penicillins and cephalosporins. ESBL-producing *K. pneumoniae* isolates containing *bla*_SHV-like_, *bla*_CTX-M-like,_ and/or *bla*_TEM-like_ genes were also reported in high percentages in other clinical isolates from Portuguese hospitals, demonstrating their high spread in this bacterial species. However, *bla*_CTX-M-like_ was found in a higher percentage than *bla*_SHV-like_ and *bla*_TEM-like_, in accordance with epidemiology data worldwide [56]. In our study, there were more samples from around 2010 than in 2020, which is why the dominance of CTX-M-types seen in Europe may not yet be reflected in the results. In fact, the prevalence of *bla*_CTX-M-like_ has been increasing considerably when compared to other ESBL types, not only in Europe but worldwide [57,58,59]. Regarding acquired AmpC β-lactamases, the low prevalence in *K. pneumoniae* agrees with other studies in the UK, Ireland, and Norway [60,61]. In Portugal, it was also detected among *Klebsiella* spp. in clinical settings, although *bla*_DHA_ in a much higher percentage than *bla*_CMY-2_ [62].

The distribution by sample types was similar to all β-lactamase genes, being detected most often in urinary samples, followed by rectal swabs, respiratory-associated infections, and blood cultures. *bla*_OXA-48-like_ was only detected in urinary isolates and respiratory-associated infections, whereas acquired AmpC β-lactamases, *bla*_DHA-like_ and *bla*_CMY-2-like_ genes, were detected in isolates from urine, respiratory-associated infections, and pus. However, it is important to notice that the number of isolates for each sample type is very different (ranging from 40 urinary isolates to 13 blood cultures), which means that for sample types with more isolates, the distribution of the genetic determinants is certainly more accurate. An analysis within each sample type revealed that *bla*_KPC-like_ is highly associated with rectal swab isolates and urine isolates (Figure 4B). This high association of KPC carbapenemases in rectal swabs has already been described in several studies [63,64,65] since *K. pneumoniae* is an intestinal pathogen subject to a huge selective pressure by antibiotic use. On the other hand, in samples from respiratory-associated infections, blood cultures, and other samples, the frequency of ESBL genes seemed higher compared to carbapenemases genes, in most cases higher than any carbapenemase gene. For example, *bla*_SHV-like_ had an almost total frequency (0.94/1), higher than *bla*_KPC-like_ (0.56/1), as well as *bla*_TEM-like_ (0.63/1) for isolates from respiratory-associated infections. Still, for all sample types, the frequency of ESBL genes was highest for *bla*_SHV-like_, followed by *bla*_TEM-like_, and finally, *bla*_CTX-M-like_.

Regarding the MDR *K. pneumoniae* isolate with no resistance genes detected, given the enormous diversity of resistance elements, this isolate might simply carry a less common resistance gene different from those tested or have another resistance mechanism (e.g., efflux pumps or porins modifications) responsible for the phenotypic pattern [66]. All other isolates tested positive for one or more resistance genes corresponding to at least one of the predicted resistance phenotypes (Appendix A). Still, not all phenotypes predicted for each isolate have been confirmed by the presence of resistance genes. As mentioned above, resistance to different β-lactams can result from the production of enzymes that directly hydrolyze these or from the combination of enzymes, such as AmpC and/or ESBL, with other cellular mechanisms, such as decreased membrane permeability to antibiotics [67]. It also needs to be considered that the development of resistance depends on the level of gene expression, and thus, it cannot be excluded that resistance genes cannot express in-vitro (susceptibility tests) but show expression *in-vivo* and/or truncated β-lactamase genes may be detected by molecular methods but not be expressed [68]. Lastly, resistance phenotypes may result from the production of β-lactam resistance genes other than those tested in this study.

In addition, approximately 80% of MDR *K. pneumoniae* isolates were found to have two or more tested resistance genes (Appendix A). This evidence may be associated with the high capacity for clonal expansion and exchange of mobile genetic elements (e.g., β-lactam resistance genes) of this species, which promotes increased resistance to antibiotics [67]. That said, the potential for resistance of this important nosocomial pathogen is alarming, and this work highlights the urge to introduce in the marketplace of new alternative therapies to fight infections due to *K. pneumoniae*.

## 4. Materials and Methods

### 4.1. Bacterial Isolates

One hundred and two *K. pneumoniae* isolates with an MDR profile were provided by the microbiology laboratories of the Hospital de Braga and of the Centro Hospitalar De Trás Os-Montes E Alto Douro, E.P.E., both in northern Portugal. Isolates were collected over a period of ten years (2010–2020). This study was performed in line with the principles of the Declaration of Helsinki and the guidelines of Good Clinical Practice. Approval was granted by the Ethics Committee of Life and Health Sciences Research Institute (ICVS), School of Health Sciences, University of Minho, Braga, Portugal. The isolates provided were previously identified using VITEK^®^2 Compact B System (bioMérieux, Marcy-l’Étoile, France). Even so, the bacterial isolates were cultured on MacConkey agar (BioLife, Milan, Italy) plates, and typical *K. pneumoniae* colonies were sub-cultured onto trypticase soy agar (TSA, Liofilchem, Roseto degli Abruzzi, Italy) for further analysis. Isolates were also preserved in 20% (*v/v*) glycerol (Riedel-de-Haën, Seelze, Germany) in tryptic soy broth (TSB, Liofilchem) and stored at −80 °C. The sample sources and isolation dates from all isolates were gathered together with the bacterial species information.

### 4.2. Antimicrobial Susceptibility Testing

Antibiograms were obtained by a VITEK 2 compact system available at the hospital laboratories. The antibiotic susceptibility profiles were then confirmed and complemented to meet the recommended list of antimicrobial agents proposed by the European Center for Disease Prevention and Control (ECDC) and the Centers for Disease Control and Prevention (CDC) [67], using the reference disk diffusion method as described by the EUCAST [68]. The list of β-lactam antibiotics tested is shown in Table 1. Quality control (QC) was performed using *Escherichia coli* ATCC 25922 and EUCAST breakpoint tables, version 12.0, were used in the interpretation of the susceptibility results [69]. Based on the antimicrobial susceptibility profiles, the *K. pneumoniae* isolates were classified as potential producers of CPN, ESBL, and/or AmpC according to the EUCAST guidelines for the detection of resistance mechanisms of clinical and/or public health importance [15]. In summary, the isolates were considered suspected of CPN-producing *Enterobacteriaceae* if the result of susceptibility to meropenem with disk diffusion was <28 mm. To be considered a potential ESBL-producing *Enterobacteriaceae*, the isolates required an intermediate or resistance result to one or both of cefotaxime (CTA) and ceftazidime (CTZ), while AmpC β-lactamase-producing *Enterobacteriaceae* required a resistant profile for CTA or CTZ, and cefoxitin (CXI). 

### 4.3. Detection of Acquired β-Lactam Resistance Genes by Multiplex PCR 

Four multiplex PCR assays for the detection of the most common β-lactam resistance genes of CPN, ESBL, or AmpC were designed using primers described in the literature. The different multiplexes were optimized using positive control strains containing the resistance genes of interest (Table 2). Subsequently, the *K. pneumoniae* isolates were screened for the presence of the acquired β-lactam resistance genes. In brief, a loop (1 µL) of each bacterial culture was homogenized in 500 µL of ultra-pure water and then boiled for 15 min. The suspension was centrifuged (12,000× *g*) for 5 min, and the supernatant (2 µL) was used as a DNA template. PCR was carried out in a 20 μL reaction mixture containing 10× reaction buffer (NzyTech, Lisbon, Portugal), 1.5 mM MgCl_2_, 0.25 mM of dNTPs (dATP, dTTP, dGTP, and dCTP), five pmol each of forward and reverse primers and 1.25 U of Supreme NZYTaq II DNA polymerase (NzyTech) in an MJ Mini Personal Thermal Cycler (Bio-Rad, CA, USA). The PCR conditions included initial denaturation at 95 °C for 5 min, followed by 30 cycles of denaturation at 94 °C for 30 s, annealing for 30 s at an optimized temperature, and extension at 72 °C for 30 s. A final extension was set at 72 °C for 5 min. The PCR products were mixed with 6x NZYDNA loading dye (NzyTech), subjected to electrophoresis at 100 V for one h in a 1.5 % (*w/v*) agarose gel, previously stained with GreenSafe Premium (NzyTech), and finally analyzed under UV light.

### 4.4. Statistical Analysis

All data were analyzed with GraphPad Prism version 7 (La Jolla, CA, USA) using one-way ANOVA with Holm-Sidak’s multiple comparisons test. Values with a *p* < 0.05 were considered statistically significant.

## 5. Conclusions

In summary, this work applied four multiplex PCRs to rapidly detect the presence of the most common acquired resistance genes of carbapenemases, ESBL, and AmpC β-lactamases. As a fast, reliable, and low-cost methodology, this can be a valuable tool in epidemiological studies as confirmation of standard phenotypic characterization methodologies. Overall, it was observed a diverse pool of β-lactam resistance genetic determinants disseminated among *K. pneumoniae* isolates (*bla*_KPC-like_, *bla*_OXA-48-like_, *bla*_TEM-like_, *bla*_CTX-M-like_, *bla*_SHV-like_, *bla*_CMY-2-like_ and *bla*_DHA-like_), revealing a diverse threat to the effectiveness of the β-lactam class. In addition, this long-term study confirmed the high spread of multiple β-lactam resistance genes among *K. pneumoniae* isolates from hospital settings in Portugal, highlighting the need for mitigation measures to prevent the advancement of this problem.

## Figures and Tables

**Figure 1 antibiotics-11-01613-f001:**
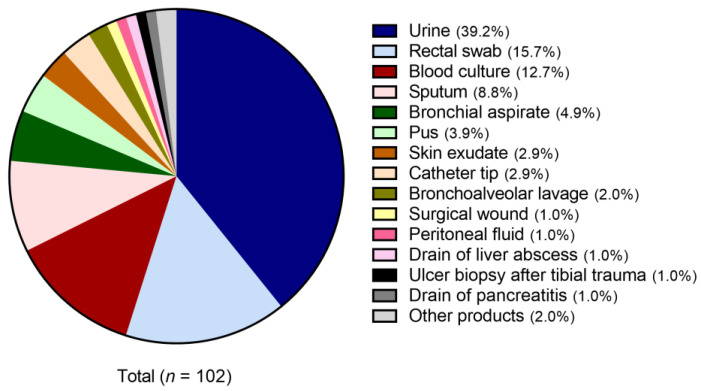
Distribution of *Klebsiella pneumoniae* isolates according to the source isolation sample.

**Figure 2 antibiotics-11-01613-f002:**
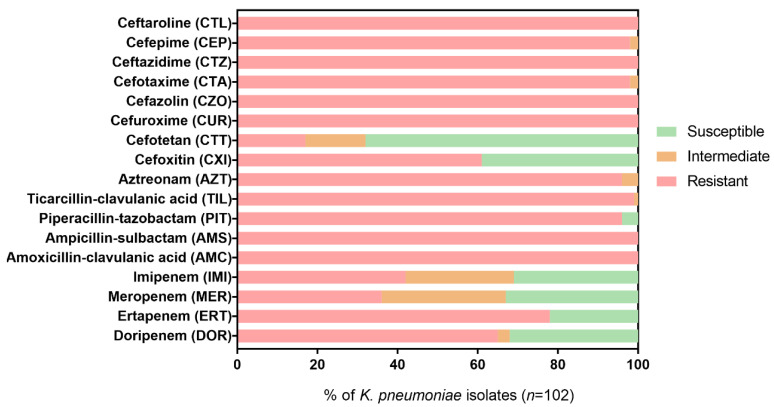
Antibiotic resistance portfolio of *K. pneumoniae* isolates determined according to EUCAST criteria. Cefotetan (CTT) is not included in EUCAST breakpoints guidelines, so we used the CLSI criteria. Note. The percentage of *K. pneumoniae* considered resistant according to the guidelines was statistically different from the % of isolates classified as intermediate or susceptible (one-way ANOVA with Holm-Sidak’s multiple comparisons tests; *p* < 0.05).

**Figure 3 antibiotics-11-01613-f003:**
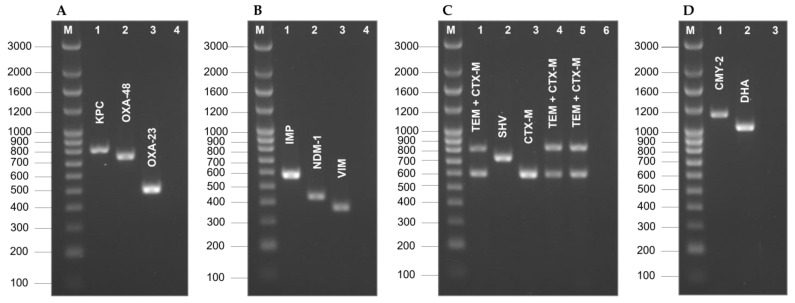
Optimization of the four multiplex PCR assays for the screening of acquired β-lactamases genes. (**A**)—Multiplex PCR assay for *bla*_KPC_/*bla*_OXA-48_/*bla*_OXA-23_ genes. Lanes: 1—*K. pneumoniae* R09 (KPC+, TEM+, SHV+); 2—*C. freundii* OXA 08 24 (OXA-48+); 3—*A. baumannii* MDU 03 21 (OXA-23+); 4—negative control (*E. col*i ATCC 25922); M, molecular size marker (in bp). (**B**)—Multiplex PCR assay for blaIMP/blaNDM-1/blaVIM genes. Lanes: 1—*P. aeruginosa* R08 (IMP+); 2—*E. coli* MDU 02 37 (NDM-1+); 3—*K. pneumoniae* MDU 02 40 (KPC-2+, VIM-1+); 4—negative control (*E. coli* ATCC 25922); M, molecular size marker (in bp). (**C**)—Multiplex PCR assay for *bla*_TEM_/*bla*_SHV_/*bla*_CTX-M_ genes. Lanes: 1—*E. coli* R02 (CTX-M+, TEM+); 2—*E. coli* H1015 (SHV-12+); 3—*E. coli* H1043 (CTX-M-I+); 4—*E. coli* H1046 (CTX-M-II+, TEM+), 5—*E. coli* H995 (CTX-M-IX+, TEM+), 6—negative control (*E. coli* ATCC 25922); M, molecular size marker (in bp). (**D**)—Multiplex PCR assay for the *bla*_TEM_/*bla*_SHV_/*bla*_CTX-M_ genes. Lanes: 1—*E. coli* C1988 (CMY-2+); 2—*K. pneumoniae* H642 (DHA-1+); 3—negative control (*E. coli* ATCC 25922); M, molecular size marker (in bp).

**Figure 4 antibiotics-11-01613-f004:**
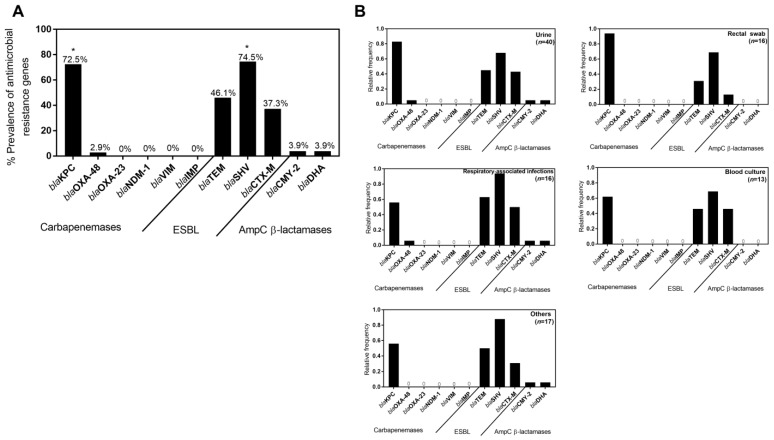
(**A**) Prevalence of acquired β-lactam resistance genes in MDR *K. pneumoniae* isolates from two Portuguese hospitals. * Prevalence of a specific resistance gene was statistically different when compared with the other genes from the same group (one-way ANOVA; *p* < 0.05). (**B**) Relative frequency of acquired β-lactam resistance genes by sample source. The isolates were categorized into five major sample sources: urine, rectal swab, respiratory-associated infections (including sputum, bronchial aspirate, and bronchoalveolar lavage isolates), blood culture, and others (including remaining sample sources).

**Table 1 antibiotics-11-01613-t001:** β-lactam antimicrobial categories and agents considered for this study according to the ECDC/CDC guideline.

Antimicrobial Category	Antimicrobial Agent (Abbreviation)
Carbapenems	Ertapenem (ERT)
	Imipenem (IMI)
	Meropenem (MER)
	Doripenem (DOR)
Penicillins + β-lactamase inhibitors	Amoxicillin-clavulanic acid (AMC)
	Ampicillin-sulbactam (AMS)
Antipseudomonal penicillins + β-lactamase inhibitors	Ticarcillin-clavulanic acid (TIL)
	Piperacillin-tazobactam (PIT)
Monobactams	Aztreonam (AZT)
Cephamycins	Cefoxitin (CXI)
	Cefotetan (CTT)
Non-extended-spectrum cephalosporins; 1st and 2nd generation cephalosporins	Cefazolin (CZO)
	Cefuroxime (CUR)
Extended-spectrum cephalosporins; 3rd and 4th generation cephalosporins	Cefotaxime (CTA)
	Ceftazidime (CTZ)
	Cefepime (CEP)
Anti-MRSA cephalosporins	Ceftaroline (CTL)

**Table 2 antibiotics-11-01613-t002:** Sequences of primers and conditions used for each multiplex PCR for the detection of resistance genes in the study.

Target Gene		Primer (5’–3’)	Amplicon Size	Annealing Temperature	Ref.
Multiplex 1 Class A and class D carbapenemases	*bla* _KPC-like_	CGTCTAGTTCTGCTGTCTTG	798 bp	60 °C	[17]
		CTTGTCATCCTTGTTAGGCG			
	*bla* _OXA-48-like_	TTGGTGGCATCGATTATCGG	743 bp		[17]
		GAGCACTTCTTTTGTGATGGC			
	*bla* _OXA-23-like_	GATCGGATTGGAGAACCAGA	501 bp		[18]
		ATTTCTGACCGCATTTCCAT			
Multiplex 2 Class B carbapenemases (MBL)	*bla* _IMP-like_	GAAGGYGTTTATGTTCATAC	587 bp	54 °C	[19]
		GTAMGTTTCAAGAGTGATGC			
	*bla* _NDM-1-like_	TAAAATACCTTGAGCGGGC	439 bp		[20]
		AAATGGAAACTGGCGACC			
	*bla* _VIM-like_	GTTTGGTCGCATATCGCAAC	382 bp		[19]
		AATGCGCAGCACCAGGATAG			
Multiplex 3 ESBL	*bla* _TEM-like_	CATTTYCGTGTCGCCCTTATTC	800 bp	60 °C	[13]
		CGTTCATCCATAGTTGCCTGAC			
	*bla* _SHV-like_	AGCCGCTTGAGCAAATTAAAC	713 bp		[21]
		ATCCCGCAGATAAATCACCAC			
	*bla* _CTX-M-like_	ATGTGCAGYACCAGTAARGTKATGGC	593 bp		[22]
		TGGGTRAARTARGTSACCAGAAYCAGCGG			
Multiplex 4 AmpC β-lactamases	*bla* _CMY-2-like_	ATGATGAAAAAATCGTTATGCT	1145 bp	60 °C	[23]
		TTATTGCAGCTTTTCAAGAATGCG			
	*bla* _DHA-like_	TGATGGCACAGCAGGATATTC	997		[13]
		GCTTTGACTCTTTCGGTATTCG			

## Data Availability

Not applicable.

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
