# Peer review of "Exploring the Antibiotic Resistance Profile of Clinical Klebsiella pneumoniae Isolates in Portugal"

_antibiotics, 2022, doi:10.3390/antibiotics11111613_

Round 1
Reviewer 1 Report
Oliveira et al. assessed the presence of genetic determinants of beta-lactam resistance in 102 multi-drug resistant (MDR) K. pneumoniae isolates under hospital settings. They found a high-rate (>90%) of resistance to most beta-lactam antibiotics, except carbapenems and cephamycins. They further developed four PCR multiplex protocols to detect the commonly observed bla genes. The manuscript is overall in its good shape. I enjoyed the neat PCR bands shown in Fig. 3A.
I have one major comment:
1. The manuscript could improve by composing a figure comparing the 11 bla targets included in Table 2. This figure can have a color scale in terms of DNA sequence similarity. In addition, for each bla gene, please mark the positions of each primer pair.
Minor comments:
1. Lines 21–22, "K. pneumoniae" → "Klebsiella pneumoniae".
2. Line 28, "23.5-34.3%" → "23.5–34.3%". Use en dash (–), not hyphen (-). Check this throughout the manuscript.
3. Lines 59–61, rewrite "Importantly … ".
Author Response
Major comment:
Point 1: The manuscript could improve by composing a figure comparing the 11 bla targets included in Table 2. This figure can have a color scale in terms of DNA sequence similarity. In addition, for each bla gene, please mark the positions of each primer pair.
Response 1: Agreed. A supplementary Figure (Figure S1) with a type-sequence of each bla target with indication of the primers binding sites is now included. The figure does not represent a sequence comparison of all bla genes as they are very different. In addition, each bla type has several variations described (most of them hundreds) so it is difficult to make a graphical representation of all variations of the 11 bla genes. Still, this figure demonstrates a representative sequence of each gene with the primer binding site in order to show the expected amplicon size in the PCR method used.
Minor comments:
Point 2: Lines 21–22, "K. pneumoniae" → "Klebsiella pneumoniae".
Response 2: Agreed!
Point 3: Line 28, "23.5-34.3%" → "23.5–34.3%". Use en dash (–), not hyphen (-). Check this throughout the manuscript.
Response 3: Agreed!
Point 4: Lines 59–61, rewrite "Importantly … ".
Response 4: The sentence was rewritten to make it easier to follow. See lines 59-61.
Reviewer 2 Report
In this manuscript of "Exploring the antibiotic resistance profile of clinical Klebsiella pneumoniae isolates in Portugal", the authors collected 102 multi-drug resistant (MDR) K. pneumoniae isolates by the laboratories from two Portuguese hospitals between 2010-2020. Four multiplex PCRs was applied to rapidly detect the presence of the most common acquired resistance genes of carbapenemases, ESBL and AmpC beta-lactamases. The study confirmed the high spread of multiple beta-lactam resistance genes among K. pneumoniae isolates from hospital in Portugal.
However, some issues should be addressed for further consideration.
1. In Figure 2, the notes on the right side are too crowd to show all the words.
2. Please make Table 1 and Figure 2 be closed to section 2.2 for easy understanding. β-lactam antimicrobial categories were mentioned several times in this section. Similar for Table 2 and figure 3A.
3. Please double check that the percentage data presented in section 2.3 from line 136 to 140 is different from the data in Figure 3B.
4. The bacteria names should be italic through the manuscript.
Author Response
Point 1: In Figure 2, the notes on the right side are too crowd to show all the words.
Response 1: Agreed. Formatting problems were fixed.
Point 2: Please make Table 1 and Figure 2 be closed to section 2.2 for easy understanding. β-lactam antimicrobial categories were mentioned several times in this section. Similar for Table 2 and figure 3A.
Response 2: Agreed. The figures and tables have been relocated in the text, closer to where they are initially mentioned.
Point 3: Please double check that the percentage data presented in section 2.3 from line 136 to 140 is different from the data in Figure 3B.
Response 3: Agreed. The reviewer was right. There was a disagreement between the percentages in the text and the figures. Everything has been reviewed and corrected. See lines 166-172.
Point 4: The bacteria names should be italic through the manuscript.
Response 4: Agreed. The entire manuscript has been revised and bacteria names put in italics.
Reviewer 3 Report
A manuscript by Oliveira et al. demonstrated the phenotype and genotype characteristics of antibiotic resistance among Klebsiella pneumoniae clinical isolates in Portugal. This study also provided an application of four multiplex PCRs for rapid detection of the most common CPNs, ESBLs, and AmpC genes. However, I have some comments to be addressed.
Major comments
1. Lines 211 - 215 and Table S1: How did the authors classify potential CPN-producing isolates with MER susceptible phenotype? MER resistance phenotype is generally used to screen CPN-producing isolates as recommended by EUCAST, while the phenotypic intermediate or resistance to at least one of carbapenem antibiotics (especially ertapenem) is commonly used to identify CPN production in Enterobacterales as recommended by CLSI. However, some carbapenem-susceptible isolates in the present study were also classified as CPN-producing isolates, as shown in Table S1. Moreover, the CPN producer seems to be classified by the presence of blaKPC gene, except for the blaKPC-positive H53 isolate. Please provide more explanation on this issue.
2. Since the K. pneumoniae clinical isolates were obtained from several sources of samples and collected over ten years (2010 - 2020), it might be better to analyze the distribution and correlation of antibiotic resistance profiles among these isolates, according to the years and sample sources.
3. In this study, the authors predicted the production of β-lactamase enzymes (CPNs, ESBLs, and AmpC), based on antibiotic resistance profiles in the K. pneumoniae clinical isolates. According to Table S1, I hypothesize that some isolates probably carry truncated β-lactamase genes that cannot be expressed and some isolates are probably resistant to the antibiotics by other mechanisms such as porin loss and overexpression of efflux pumps. Additionally, the different classes of these enzymes can also provide various antibiotic resistance profiles in the studied isolates. Many methods are then recommended to detect these enzyme productions, which could be performed by comparing the susceptibility of β-lactam with β-lactam + β-lactamase inhibitor (combined disk test). Thus, the limitation or suggestion of further investigation of β-lactamase production could be also mentioned.
- Epidemiology should be discussed and provide comparative analysis with other area in European country.
Minor comments
1. Please check the italic of bacterial and gene names throughout the manuscript.
2. Line 76: “… or carbapenems, and are poorly inhibited …” – “… or carbapenems, are poorly inhibited …”
3. Lines 102, 108, 132, 178, 185, 215, 231, and 244: “Fig.” – “Figure”
4. Lines 108 - 113 and 204: The full names of “ERT”, “IMI”, “MER”, “DOR”, “CXI”, “CTT”, “CTA”, and “CTZ” could be also provided.
5. Lines 170 - 172: “ESCAPE” – did the authors mean “ESKAPE (Enterococcus faecium, Staphylococcus aureus, Klebsiella pneumoniae, Acinetobacter baumannii, Pseudomonas aeruginosa, and Enterobacter spp.)?
6. Line 225: “KPC genes” could be replaced by “blaKPC gene”
7. Lines 231 - 232: “VIM-like and NDM-1 genes” could be replaced by “blaVIM-like and blaNDM-1 genes”.
8. Lines 233 - 234: “SHV-like, CTX-M-like and/or TEM-like genes” could be replaced by “blaSHV-like, blaCTX-M-like, and/or blaTEM-like genes”.
9. Line 236: “DHA-like and CMY-2-like genes” could be replaced by “blaDHA-like and blaCMY-2-like genes”.
1- Lines 237 and 238: “DHA” and “CMY-2” could be replaced by “blaDHA” and “blaCMY-2”, respectively.
1- Line 254: “resistance genes beta-lactams” – it might be better to rephrase it as “beta-lactams resistance genes”.
1- Lines 325 - 326: “(KPC, OXA-48, TEM, CTX-M, SHV, CMY-2 and DHA)” could be replaced by “(blaKPC, blaOXA-48, blaTEM, blaCTX-M, blaSHV, blaCMY-2 and blaDHA)”.
Author Response
Major comments
Point 1: Lines 211 - 215 and Table S1: How did the authors classify potential CPN-producing isolates with MER susceptible phenotype? MER resistance phenotype is generally used to screen CPN-producing isolates as recommended by EUCAST, while the phenotypic intermediate or resistance to at least one of carbapenem antibiotics (especially ertapenem) is commonly used to identify CPN production in Enterobacterales as recommended by CLSI. However, some carbapenem-susceptible isolates in the present study were also classified as CPN-producing isolates, as shown in Table S1. Moreover, the CPN producer seems to be classified by the presence of blaKPC gene, except for the blaKPC-positive H53 isolate. Please provide more explanation on this issue.
Response 1: We agreed that this statement can be confusing. However according to the latest version of the EUCAST guidelines for detection of resistance mechanisms and specific resistances of clinical and/or epidemiological importance, a phenotype for meropenem <28 mm with disk diffusion (or MIC >0.125 mg/L) in all Enterobacteriaceae should be considered a potential carrier of a carbapenemases gene (with some exceptions). On the other hand, the S breakpoint for meropenem according to EUCAST is ≥22 mm. Therefore, there is a range between 22-28 mm in the disc diffusion result which, despite showing a susceptible phenotype for MER, should be considered potential carrier of carbapenemases. In our case, we had some isolates with MER susceptibility phenotype (between 22-25mm) that revealed blaKPC gene. For isolate H53 there is an error in table S1 that has now been corrected. This isolate is one of those cases, although it has a MER susceptible phenotype (27 mm) it was considered a potential producer of carbapenemases and the genotypic characterization revealed the presence of blaKPC. See lines 253-267.
Point 2: Since the K. pneumoniae clinical isolates were obtained from several sources of samples and collected over ten years (2010 - 2020), it might be better to analyze the distribution and correlation of antibiotic resistance profiles among these isolates, according to the years and sample sources.
Response 2: We agree that analysis by years and sample types would improve the analysis presented. Therefore, analysis by sample type was introduced (Figure 4B) and a discussion was held. We have divided those isolates into 5 groups according to the samples source. Regarding the evaluation over time, we did not perform the analysis by years, since the distribution of samples is not uniform across years. See figure 4B and lines 193-202, 299-311.
Point 3: In this study, the authors predicted the production of β-lactamase enzymes (CPNs, ESBLs, and AmpC), based on antibiotic resistance profiles in the K. pneumoniae clinical isolates. According to Table S1, I hypothesize that some isolates probably carry truncated β-lactamase genes that cannot be expressed and some isolates are probably resistant to the antibiotics by other mechanisms such as porin loss and overexpression of efflux pumps. Additionally, the different classes of these enzymes can also provide various antibiotic resistance profiles in the studied isolates. Many methods are then recommended to detect these enzyme productions, which could be performed by comparing the susceptibility of β-lactam with β-lactam + β-lactamase inhibitor (combined disk test). Thus, the limitation or suggestion of further investigation of β-lactamase production could be also mentioned.
Response 3: We agree. The hypothesis of the influence of the expression of resistance genes on the demonstrated phenotype was introduced in the discussion, and the possibility of alternative resistance mechanisms was highlighted. Furthermore, it was mentioned the possibility that other phenotyping methods (combined disk test, etc.) could improve the prediction of resistance. Please see lines 322-327.
Point 4: Epidemiology should be discussed and provide comparative analysis with other area in European country.
Response 4: Agreed. Comparison of the prevalence of resistance genes with other European studies was introduced. Please see lines 273-275, 292-294.
Minor comments
Point 5: Please check the italic of bacterial and gene names throughout the manuscript.
Response 5: The entire manuscript has been revised and bacteria and gene names put in italics.
Point 6: Line 76: “… or carbapenems, and are poorly inhibited …” – “… or carbapenems, are poorly inhibited …”
Response 6: Agreed. We have rewritten the statement.
Point 7: Lines 102, 108, 132, 178, 185, 215, 231, and 244: “Fig.” – “Figure”
Response 7: Done.
Point 8: Lines 108 - 113 and 204: The full names of “ERT”, “IMI”, “MER”, “DOR”, “CXI”, “CTT”, “CTA”, and “CTZ” could be also provided.
Response 8: We have introduced the full names of the antimicrobial agents in the first appearance in the main text.
Point 9: Lines 170 - 172: “ESCAPE” – did the authors mean “ESKAPE (Enterococcus faecium, Staphylococcus aureus, Klebsiella pneumoniae, Acinetobacter baumannii, Pseudomonas aeruginosa, and Enterobacter spp.)?
Response 9: Yes, correction was made.
Point 10: Line 225: “KPC genes” could be replaced by “blaKPC gene”
Response 10: Agreed.
Point 11: Lines 231 - 232: “VIM-like and NDM-1 genes” could be replaced by “blaVIM-like and blaNDM-1 genes”.
Response 11: Agreed.
Point 12: Lines 233 - 234: “SHV-like, CTX-M-like and/or TEM-like genes” could be replaced by “blaSHV-like, blaCTX-M-like, and/or blaTEM-like genes”.
Response 12: Agreed.
Point 13: Line 236: “DHA-like and CMY-2-like genes” could be replaced by “blaDHA-like and blaCMY-2-like genes”.
Response 13: Agreed.
Point 14: Lines 237 and 238: “DHA” and “CMY-2” could be replaced by “blaDHA” and “blaCMY-2”, respectively.
Response 14: Agreed.
Point 15: Line 254: “resistance genes beta-lactams” – it might be better to rephrase it as “beta-lactams resistance genes”.
Response 15: Agreed.
Point 16: Lines 325 - 326: “(KPC, OXA-48, TEM, CTX-M, SHV, CMY-2 and DHA)” could be replaced by “(blaKPC, blaOXA-48, blaTEM, blaCTX-M, blaSHV, blaCMY-2 and blaDHA)”.
Response 16: Agreed.
Round 2
Reviewer 1 Report
Thank you for making the revisions. I have no further questions.
Author Response
Thanks for your suggestions!
Reviewer 2 Report
The authors have done all the necessary correction and now the manuscript could be accepted in this current version.
Author Response
Thanks for your suggestions!
Reviewer 3 Report
The authors addressed all of my questions in this revised MS. However, the Epidemiology aspects need to be further analyzed and discussed in deep.
Author Response
Point 1: The authors addressed all of my questions in this revised MS. However, the Epidemiology aspects need to be further analyzed and discussed in deep.
Response 1: We have introduced more epidemiological data from Europe, including Portugal, and other parts of the world. A more comprehensive comparison of our results with the data from these studies was made. See lines 280-290, 297-300, 312-318.
Round 3
Reviewer 3 Report
-